# Sample Size Effects on Petrophysical Characterization and Fluid-to-Pore Accessibility of Natural Rocks

**DOI:** 10.3390/nano13101651

**Published:** 2023-05-16

**Authors:** Qiming Wang, Qinhong Hu, Chen Zhao, Yang Wang, Tao Zhang, Jan Ilavsky, Mengdi Sun, Linhao Zhang, Yi Shu

**Affiliations:** 1Shandong Provincial Key Laboratory of Deep Oil and Gas, China University of Petroleum (East China), Qingdao 266580, China; 20230014@upc.edu.cn (Q.W.);; 2Department of Earth and Environmental Sciences, The University of Texas at Arlington, Arlington, TX 76019, USA; 3School of Earth Science and Resources, Chang’an University, Xi’an 710054, China; 4X-ray Science Division, Advanced Photon Source, Argonne National Laboratory, Lemont, IL 60439, USA; 5Key Laboratory of Continental Shale Hydrocarbon Accumulation and Efficient Development, Ministry of Education, Northeast Petroleum University, Daqing 163318, China; 6Key Laboratory of Tectonics and Petroleum Resources of Ministry of Education, China University of Geosciences, Wuhan 430074, China; 7Petroleum Exploration and Development, Jianghan Oilfield Branch Company, Sinopec, Wuhan 430223, China; shuy24185.jhyt@sinopec.com

**Keywords:** particle size effect, petrophysical properties, mercury intrusion porosimetry, X-ray scattering, particle density

## Abstract

Laboratory-scale analysis of natural rocks provides petrophysical properties such as density, porosity, pore diameter/pore-throat diameter distribution, and fluid accessibility, in addition to the size and shape of framework grains and their contact relationship with the rock matrix. Different types of laboratory approaches for petrophysical characterization involve the use of a range of sample sizes. While the sample sizes selected should aim to be representative of the rock body, there are inherent limitations imposed by the analytical principles and holding capacities of the different experimental apparatuses, with many instruments only able to accept samples at the μm–mm scale. Therefore, a total of nine (three limestones, three shales, two sandstones, and one dolomite) samples were collected from Texas to fill the knowledge gap of the sample size effect on the resultant petrophysical characteristics. The sample sizes ranged from 3 cm cubes to <75 μm particles. Using a combination of petrographic microscopy, helium expansion pycnometry, water immersion porosimetry, mercury intrusion porosimetry, and (ultra-) small-angle X-ray scattering, the impact of sample size on the petrophysical properties of these samples was systematically investigated here. The results suggest that the sample size effect is influenced by both pore structure changes during crushing and sample size-dependent fluid-to-pore connectivity.

## 1. Introduction

Laboratory-scale petrophysical analyses of natural rocks help engineers and geologists in their evaluation of petroleum recovery, coal mining, CO_2_ sequestration and utilization, nuclear waste repository, groundwater protection, and geothermal energy exploration (e.g., [1,2,3,4,5,6,7]). Because of the complexities of mineral composition and pore structure (with regards to both geometry and topology; [1,8]) in natural rocks, there is a lingering debate over the most reliable and reproducible laboratory protocols to be used in petrophysical characterization. Furthermore, the limitation of the sample holding capacity of various experimental apparatuses leads to concerns over to what degree the small sample size is representative of the rock body as a whole. Large-sized samples, such as 10 cm sized full-diameter cores, are somewhat more representative of the heterogeneity of the rock body than sub-centimeter-sized samples such as cuttings, but routine methodologies for appropriate sample sizes barely exist for direct petrophysical analyses [9]. Smaller samples at various sizes (e.g., 2.54 cm diameter core plugs, cuttings, or crushed rocks) are more commonly used, but the results collected from differently sized samples can vary substantially [10,11,12].

The sample size effect (i.e., a variation in test results due to different sample sizes) on measured rock properties is even more challenging for tight rocks with nano-darcy permeability. For example, routine core plug analysis for permeability is not practical because a probing gas will not penetrate the cm sized cylindrical cores in sufficient volume and in a reasonable time [10]. To address this issue, Luffel and Guidry [10] developed a crushed sample method to obtain porosity, by analyzing porosity in shale and sandstone at several different sample sizes. Luffel et al. [13] introduced the crushed sample (at hundreds μm in sizes) method to measure extremely low (nano-darcy level) matrix permeability for shale, which is also widely known as the GRI (Gas Research Institute) matrix permeability method. However, several researchers [14,15,16,17,18] suggested that the permeability results from the GRI approach are highly variable for the same rock sample at different sizes. Several researchers also noted that the measured porosity using mercury intrusion porosimetry (MIP) for various crushed shales is sample size-dependent [11,14,19,20,21]. The gas physisorption method has also been used to analyze the relationship between pore structure and sample size [22,23,24,25]. Liu et al. [26] suggests that the process of coal and shale crushing alters the pore shapes, and also influences porosity, pore size distribution, and specific surface areas. From previous studies, there are three major, and sometimes conflicting, conclusions with respect to the impact of sample size reduction such as crushing: (1) stimulation of artificial microfractures [16,20]; (2) opening of isolated pores [24,27,28]; and (3) elimination of pre-existing pores and microfractures [20,29]. However, to what degree the crushing influences the petrophysical characterization results, and the mechanisms of the resulting variations, are not well understood, and how crushing changes the framework grains and sedimentary texture is usually ignored.

The influences of the crushing process and sample size effect on petrophysical test results have been noticed but have not been synthetically investigated yet. It is critical to investigate these unknowns to improve the representativeness of, and confidence in, data from laboratory analyses of various rock types. In this study, nine samples from Texas (Atco Chalk, Eagle Ford B calcareous shale, Eagle Ford A dolomitic ash bed, Buda limestone, Salmon Peak limestone, Woodbine Sandstone, Paluxy Sandstone, Haynesville Shale, and Barnett Shale) were selected to represent several different lithologies of rock formations. Varying mineral compositions and depositional environments result in different petrophysical properties of these rock samples. Eight different sample sizes were generated per sample, ranging from 3 cm cubes to <75 μm particles. An integrated approach, combining X-ray diffraction (XRD), petrographic microscopy, helium expansion pycnometry (HEP), water immersion porosimetry (WIP), MIP, and (ultra-) small-angle X-ray scattering [(U)SAXS], was used to determine the sample size effect on petrophysical properties such as porosity, pore-throat distribution, pore diameter distribution, and connectivity. A conclusion based on the surface zone/interior zone is proposed to explain the variation in results as a function of sample size.

## 2. Samples and Methods

### 2.1. Sample Collection and Preparation

A range of samples were collected across Texas (Figure 1). Atco Chalk, Eagle Ford B calcareous Shale, Eagle Ford A dolomitic ash bed, Buda Limestone, and Salmon Peak Limestone were collected from road cuts from Del Rio city to past Langtry along Highway 90. Woodbine Sandstone was collected from an outcrop near Grapevine Lake in the Dallas–Fort Worth area. Paluxy Sandstone was collected from an outcrop near Glen Rose city. Haynesville Shale has no outcrop exposed, and its core samples were collected from a well in San Augustine County. The Barnett Shale was collected from an outcrop in a quarry near San Saba City. In addition, a quartz crystal sample of several centimeters in diameter (Godfrey, ON, Canada) from Ward’s Natural Science with pure mineralogy and nearly zero porosity was included for various validation purposes (as a “standard”), such as subjection to the same sample size reduction process, data processing from MIP to consider the conformance effect, and investigation of the potential influence of sample size on the particle density results from HEP approach. The samples were first cut into 3, 2, and 1 cm cubes using a circular saw (6″ Lapidary Trim Saw, Kingsley North) with a minimal use of water; then, the rest of the rock fragments were crushed by using a mortar and pestle down to 1.70–2.38 mm (8–12 mesh), 500–841 μm (20–35 mesh), 177–500 μm (35–80 mesh), 75–177 μm (80–200 mesh), and <75 μm. During the crushing process, the small pieces (<2.38 mm) were split from the large chunk of the rock after brief use of the pestle, and then transferred to the sieve stack to minimize the influence of the crushing process on rocks. The remaining pieces > 2.38 mm were subjected to further crushing. A 0.8 mm wafer was also sliced off from the rock fragments for (U)SAXS analysis. The <75 μm sized sample was used only for the XRD analyses. All the samples were washed with de-ionized water to remove the dust on sample surfaces generated during sample cutting and grinding. After washing, all samples were oven-dried at 60 °C for 48 h to remove the moisture in connected pores, then cooled down to room temperature in a desiccator to minimize the moisture uptake before analyses. As multiple sample sizes and analytical methods have been used in this study, Table 1 lists all the sample sizes and the associated methods used on those sizes.

### 2.2. X-ray Diffraction (XRD)

Mineral compositions were identified and quantified using the Shimadzu MaximaX XRD-7000 diffractometer using samples with a size < 75 μm with a scanning rotation from 2° to 70°. The mineral compositions and weight percentages for the sample were obtained through spectrum analysis and whole-pattern fitting using the Minerals relational database.

### 2.3. Petrographic Microscopy

Both intact (2 cm cubes) and particle (500–841 μm, 177–500 μm, and 75–177 μm) samples were impregnated with blue epoxy, then ground down to 30 μm in thickness to make thin sections. Thin section images were captured by using a Leica 750P polarizing microscope. Mineral type, framework grains shape, and sedimentary texture were studied in detail from the intact sample thin sections. Thin sections prepared from crushed samples were used to compare the crushed with the intact samples to investigate the effect of crushing on the framework grains and sedimentary texture.

### 2.4. Helium Expansion Pycnometry (HEP)

HEP is a gas displacement-based standard method to rapidly and precisely determine the particle density of porous materials at a wide range (up to 6–7 orders of magnitude) of sample sizes. Particle density is commonly defined as the density of rock without considering the pore space. Considering the possible presence of isolated pores in the rock matrix, the definition of particle density in this study is modified to be the density of rock including the pores inaccessible to a probing fluid (e.g., helium) applied on the sample exterior. Analyses were carried out on an AccuPyc II 1345 (Micromeritics). Particle density measurements were made on samples of sizes 1 cm cube, 1.70–2.38 mm, 500–841 μm, 177–500 μm, and 75–177 μm [30]. Differences in the measured particle density can help us to assess the presence and magnitude of pores inaccessible to the fluid applied.

### 2.5. Water Immersion Porosimetry (WIP)

WIP is a liquid-based edge-accessible porosity test using a custom-designed apparatus. This method has the capability of testing various sizes of samples from the full-sized core (4 or 10.2 cm in diameter) to rock pieces of about 0.2 cm in length and width. In this study, samples of 3 cm, 2 cm, and 1 cm cubes were analyzed to examine the size effect on the effective (mercury-accessible from the sample surface) porosity. Briefly, the sample chamber was first evacuated to 0.05 torr (99.993% vacuum) and held there for 12 h to remove the air from edge-accessible pores inside the sample, flushed with CO_2_ to replace any remaining air, and evacuated for another 12 h. Boiled and cooled de-ionized water (DIW) was then introduced into the sample chamber until the samples were submerged, after which CO_2_ at 30 psi was applied in the chamber to allow DIW to further invade the pore space. After DIW saturation for 12 h, the samples were weighed in air and then immersed in DIW to determine the bulk volume by using liquid displacement based on Archimedes’ principle. From these measurements, the bulk density, particle density, and porosity of various sample sizes can be calculated [12].

### 2.6. Mercury Intrusion Porosimetry (MIP)

MIP is a common method to analyze a range of pore structure parameters such as density, porosity and pore-throat size distribution of natural rocks. The Micrometrics AutoPore 9520 was used to analyze intact and crushed samples of 1 cm^3^ cubes, 1.70–2.38 mm, 500–841 μm, 177–500 μm, and 75–177 μm. The Washburn equation [31] describes the relationship between pressure and the pore-throat diameter being invaded. Quartz samples of 1.70–2.38 mm, 500–841 μm, 177–500 μm, and 75–177 μm were also utilized to evaluate and correct the conformance effect on crushed samples.

### 2.7. (Ultra-) Small-Angle X-ray Scattering [(U)SAXS]

(U)SAXS measurements were conducted on the Beamline 9ID of Argonne National Laboratory, IL, USA. The beam area of USAXS is 0.8 mm × 0.8 mm and the beam area of SAXS is 0.2 mm × 0.8 mm [32,33]. Sample forms included 0.8mm thick rock wafers as well as crushed samples (500–841 μm, 177–500 μm, and 75–177 μm). The data processing followed the procedure of [34], with the pore size distribution calculated with the IRENA macro for Igor Pro.

## 3. Results

### 3.1. Mineral Composition and Petrographic Observation of Thin Section Samples

In this study, three terms are used, namely, framework grains, matrix, and particles. Framework grains are mineral crystals or fossils whose sizes are identifiable under the polarizing microscope. The matrix is the remaining parts of the rock whose sizes cannot be identified under the polarizing microscope, and they usually glue the framework grains together. The term particle refers to the crushed rock samples, for example, 500–841 μm particles, which could contain 100 s to 1000 s framework grains.

The mineral compositions of nine samples are presented in Table 2. Overall, the Atco Chalk, Eagle Ford A dolomitic ash bed, Eagle Ford B calcareous shale, Buda Limestone, and Salmon Peak Limestone contain carbonate content >80% by weight. Woodbine Sandstone contains 91.8% quartz and 8.2% goethite, and Paluxy Sandstone has 42.5% quartz and 57.5% ankerite. Haynesville Shale contains 25.1% quartz, 48.2% calcite, and 15.8 clay minerals, whereas the Barnett Shale contains 40.7% quartz and 41.7% clay minerals. Figure 2A1,B1,C1,D1,E1 and Figure 3A1,B2,C1,D1 show the original framework grain shape and sedimentary texture based on microscopy of thin sections. The Atco Chalk is a bioclastic packstone with 99% calcite. In the thin section images, fossils are supported by the matrix. Fossil sizes range from 0.02 to 0.5 mm, while most of them are concentrated in the range of 0.02–0.2 mm. The black-colored Eagle Ford B calcareous shale is mainly composed of calcite, with its thin-section results showing enriched calcispheres which range from 0.1 to 0.4 mm in size. The Eagle Ford A dolomitic ash bed contains euhedral to subhedral dolomite (ankerite and kutnohorite) grains with sizes ranging from 0.05 to 0.1 mm. The Buda Limestone contains mm to cm sized fossils that can be seen in the muddy matrix. Salmon Peak Limestone is a matrix-supported bioclastic limestone, and the fossils are mainly skeletal fragments of shells with sizes mostly varying from 0.1 to 5 mm, while some large fossils can reach 1 cm. The Woodbine Sandstone is very porous with grain sizes ranging from 250 to 35 μm (fine sand to coarse silt), and the pore diameters are similar to the grain sizes. Paluxy Sandstone is less porous than the Woodbine Sandstone, as it has larger grain sizes which range from 320 to 40 μm (medium sand to coarse silt); in addition, the quartz grains are cemented with ankerite matrix. Both Haynesville and Barnett Shales are dark in color and contain some fossils. In the Barnett Shale, μm scale fractures are visible in petrographic images.

The photomicrographs for three sizes of crushed samples (500–841 μm, 177–500 μm, and 75–177 μm) of the nine samples and quartz “standard” are shown in Figure 2A2–A4,B2–B4,C2–C4,D2–D4,E2–E4 and Figure 3A2–A4,B2–B4,C2–C4,D2–D4,E2–E4. For porous rocks, the crushing process leads to changes in the sedimentary texture, namely, the size and shape of framework grain, as well as the contact relationship between the matrix and framework grains. During the crushing and milling, the mechanical breakdown of rocks created artificial microfractures in some samples (Figure 2A2,B2,D2,D3,E2,E3; Figure 3D3). In the 2D view, microfractures could cut through the particles. When the original framework grain sizes are larger than, or close to, the particle size, the framework grains are broken into smaller sizes (e.g., Figure 2B2,B3,D2,D4; Figure 3C2–C4). Moreover, some framework grains are observed to be separated from the matrix or cement (Figure 2A4,B4,C4). In Figure 2C4, only dolomite grains are present, as the matrix and cement barely exist. In the Woodbine and Paluxy Sandstone samples, pores of tens to hundreds μm in the intact rock are gradually liberated with decreasing sample size. Quartz is a non-porous mineral and only breaks into small-sized particles during crushing (Figure 3E1–E4). Overall, the crushing process of porous rocks not only leads to a decrease in particle sizes but also (1) stimulates artificial microfractures, (2) breaks up framework grains, and (3) separates the framework grains from the matrix or cement.

### 3.2. Sample Size-Dependent Particle Density from HEP

A bar diagram of the particle density vs. sample size for the nine rock samples and quartz is shown in Figure 4. As the sample size decreases, quartz shows a constant particle density at ~2.65 g/cm^3^, which indicates quartz is non-porous and its structure does not change with varying sample sizes. Woodbine Sandstone also shows a constant particle density (~2.70 g/cm^3^), indicating that the pore spaces are easy to access, and the sample size reduction does not influence the results. In contrast, the other porous rocks show very different behaviors when compared to quartz. They show particle density increases with sample size decreasing.

### 3.3. WIP Analyses of Samples at the Centimeter Scale

A compilation of the WIP results (Table 3) shows the porosity vs. sample size (3, 2, and 1 cm^3^ cube) for the nine rock samples. For most of the natural rocks, the porosity is relatively consistent for different sample sizes. The exceptions are the shales. The porosity of the Eagle Ford B sample decreases from 4.27% to 1.37% as sample size decreases. In the 3 cm^3^ cubed samples of the Eagle Ford B, some visible fractures exist in samples to provide extra pore space and lead to a higher porosity. The 2 cm^3^ and 1 cm^3^ cubed samples were cut from the region without visible fractures. Both Haynesville and Barnett Shales show a gradual increase in porosity with decreasing sample sizes from 3 cm^3^ to 1 cm^3^ cubes.

### 3.4. Conformance Effect and MIP Results

The conformance effect, also called the sample enveloping issue, occurs in the MIP tests for all samples, but especially crushed samples. Before the pressurized mercury intrusion starts, mercury can incompletely envelop the crushed samples and leave interparticle pores unfilled. Therefore, porosity will be overestimated due to the presence of these interparticle pores. Due to the large influence on the porosity of rock samples, the conformance effect has been reported and corrected in previous studies [11,19,35,36,37,38]. Those studies used mathematical and physical models to eliminate the conformance effect on the MIP results. In this study, we used an experimental method to investigate and correct the conformance effect. Crushed quartz samples at sizes of 1.70–2.38 mm, 500–841 μm, 177–500 μm, and 75–177 μm were used as non-porous reference materials. When mercury is pressured into the penetrometer only filled with the crushed quartz, it can only detect the interparticle pores and reflect at what range the conformance effect will start and end. By analyzing the mercury intrusion behaviors of different sample sizes, the conformance effects on 1.70–2.38 mm, 500–841 μm, and 177–500 μm are observed in the regions where pore-throat diameters are greater than 12 μm (corresponding to a pressure of 20 psi), as compared to 6 μm (40 psi) for smaller-sized (75–177 μm) particles (Figure 5). To correct the conformance effect, the intrusion at that range is trimmed off. After such a correction, the porosity of crushed quartz of all sizes is less than 0.22%. It is worth noting that these small “apparent” porosity values may be caused by the surface roughness of the crushed quartz, manifested as a small amount of mercury intrusion signal in MIP tests. After the conformance correction was verified from quartz “standards”, the same trimming pressure for the respective particle size was applied for eight of the rock samples, except for the Woodbine Sandstone at sizes from 1 cm^3^ cube to 177–500 μm. Because the major pore-throat size of Woodbine Sandstone is larger than 10 μm, the conformance has less influence on it than other rocks (Figure 6F). If our correction is applied to Woodbine Sandstone, the information of the actual major pore systems would have been trimmed off as well. However, the 75–177 μm Woodbine Sandstone sample shows conformance overlapped its major pore system (>10 μm) and the corrected porosity results are shown in Table 3, and the pore-throat distributions from MIP runs are presented in Figure 6.

Before the conformance correction, measured porosities show different behaviors with decreasing sample sizes (Table 3). From 1 cm^3^ cube to 177–500 μm, the porosities of the Atco Chalk, Eagle Ford B, Buda Limestone, and Salmon peak Limestone samples increase with decreasing sample sizes, whereas Eagle Ford A and Woodbine Sandstone show decreasing porosity values, and Paluxy Sandstone and Haynesville Shale do not show an obvious trend. The 75–177 μm sized samples show a dramatic increase in porosity (25.4–47.23%) in all the sample sizes. Figure 6 displays the pore-throat diameter vs. incremental intrusion for cubes and particle samples. The pore-throat size distribution of cubed samples represents the original state without the influence of crushing. Samples of Atco Chalk, Eagle Ford A, Salmon Peak Limestone, and Paluxy Sandstone show a major peak in the pore-throat diameter range of 0.1–1 μm (Figure 6A,C,E), whereas the Buda Limestone has a major peak in the range of 0.1–0.01 μm (Figure 6D). The major pore-throat size of the Woodbine Sandstone is larger than 10 μm. The Eagle Ford B and Haynesville Shale show major peaks of high intrusion volumes between 0.02 and 0.002 μm (20 to 2 nm), and the Barnett Shale exhibits a broader range of major peaks at 0.3 to 0.002 μm. Overall, the most obvious intrusion differences between differently sized samples for the nine natural rocks occur for pore-throat diameters in the region of 10 to 0.01 μm.

### 3.5. (U)SAXS Analyses

As a non-destructive method, (U)SAXS has the advantage of detecting both edge-accessible and -inaccessible (internally “isolated”) pores, and thus serves as a unique tool to tease out the proportion of edge-inaccessible pores not accounted for from the fluid-intrusion approaches such as HEP, WIP, and MIP. (U)SAXS can measure pore diameters in the 1–1000 nm range and hence determine the influence of the crushing process on nm sized pores. Figure 7 shows the pore diameter vs. incremental porosity (fraction) of 0.8 mm thick wafers (non-crushed) as well as particles of sizes 500–841 μm, 177–500 μm, and 75–177 μm of all nine rock samples. The curves do not show any changes as a function of particle size, which indicates that the pore structure in the crushed samples remains largely intact over this wide pore diameter range.

## 4. Discussion

### 4.1. Influences of Crushing Process on Microfractures, Pores, and Framework Grains

As described in the sample preparation section, the samples were hand-crushed with a mortar and pestle. Previous studies have reported SEM-observed changes in the rocks induced by the crushing and milling process [17,18,20]. In their SEM results, the apertures of microfractures were around 0.1–1 μm and 3–5 μm in length with a sharp surface; all of them suggested those microfractures were induced by the crushing process, but without comparing them to the intact samples. It is possible that some of the sharp surface microfractures are artificially induced by the crushing. However, the microfractures may also be generated by sample desiccation, pressure release, or even the SEM sample preparation process [6,39]. Technically, it is difficult to determine the origin of the fractures, and there are no independent methods in the literature to investigate the probable sources of such fractures. In the thin section petrographic studies of four sample sizes from the nine rock samples and quartz reported here, it was observed that the microfractures are induced by the crushing process. However, our results indicate that the apertures of those microfractures are greater than 10 μm and the lengths are 200–500 μm (Figure 2A2,A3,B2,D2,D3,E3; Figure 3D1,D2). The intact samples do not show microfractures in the thin section images, which suggests that the crushing process likely induces artificial microfractures in the tens to hundreds μm scale in natural rocks with different lithologies.

“Isolated” pores present in natural rocks are not absolutely isolated, but rather are not easily measurable due to their small size (typically at sub-10 nm), testing duration, and resulting experimental/instrumental limitations [30]. An opening of “isolated” pores is commonly used to explain the sample size effect on rock properties [11,20,26,40]. When the sample is crushed, the specific surface area will likely increase and more pores become accessible. Considering the limitation of 2D visualization methods, such as polarizing microscopy and SEM, it is difficult to observe the pore opening experimentally. One method to quantitatively obtain the ratio of accessible to “isolated” pores is (U)SANS [41,42].

As the previously mentioned studies were more focused on microfractures stimulation and pores opening during the crushing process, the changes in framework grains during the crushing process received less attention. As shown in Figure 2A4,B2–B4 and Figure 3C2,C4, framework grains split from the matrix during crushing and break into smaller pieces than their original size. In the shale and carbonate rocks, this size reduction might not change the pore structure, because their primary particle sizes are larger than the major pore sizes. The Woodbine Sandstone and Paluxy Sandstone samples (Figure 3A1,B1) contain many pores with sizes ranging from ~50 to 500 μm. When the particle size is reduced to 177–500 μm, some pores in the ~50–500 μm range likely become accessible. When the particles are crushed to 75–177 μm, pores larger than 177 μm will not exist and pores with sizes ranging from 50 to 177 μm will be partially accessible.

### 4.2. Sample Size Effect on Pore/Pore-Throat Diameter and Its Implications

MIP has been used to determine the pore-throat diameter of porous materials. When comparing the pore-throat diameter distribution from MIP and pore diameter distribution from (U)SAXS, the relationship between sample size and pore diameter distribution is different. For five differently sized samples, the MIP results show that the mercury intrusion curves have different shapes for pore-throat diameters > 10 nm, while the pores between 2.8 and 10 nm do not show obvious changes as a function of sample size (Figure 6). The (U)SAXS results show that the sample size does not influence the pore diameter distribution from 1000 to 1 nm in all samples (Figure 7). The good agreement in pore diameter distribution curves from the intact wafer and three differently sized crushed samples (500–841, 177–500, and 75–177 μm) indicates that our crushing process does not influence the pore structure under 1000 nm for these nine natural rocks. The MIP and (U)SAXS results show a conflicting trend at the overlapping range of pore/pore-throat diameter < 1000 nm, which leads to a question about whether the differently sized samples made by crushing will change the pore structure. In previous MIP studies of different sample sizes, differences in porosity and pore-throat size distribution have been observed [11,20,40]. Comsky et al. [11] ascribed the difference in porosity and pore-throat size distribution to the conformance effect at low pressures, compressibility at high pressure, and increase in pore accessibility with decreasing sample size. Tinni et al. [20] pointed out that the porosity and the surface area are a function of the particle size and reflect the pore accessibility variation between differently sized samples. Shu et al. [40] stated that the crushing process will increase the pore connectivity by opening the isolated pores and generating artificial fractures. There are no previous (U)SAXS studies on the sample size effect on pore structure. However, similar studies have utilized (U)SANS techniques [20,43]. Zhang et al. [43] reported that the pore structure does not show a significant change or damage, but Shu et al. [20] stated that the pore volume will increase as sample size decrease. Both of their data show similar behaviors but are interpreted in opposite directions. Their data show good overlapping between sample sizes, and the difference appears on the (U)SANS curve mainly resulting from their log-scale *Y*-axis, which will exaggerate the difference at a first sight. Compared to the (U)SAXS data in this study with simultaneous analyses of both intact and particle samples, both (U)SANS and (U)SAXS methods show similar results, whereby the pore diameter distribution of the measurable nm range is barely changed during the crushing.

The methods for measuring petrophysical properties of rocks can be divided into two groups, namely, (i) fluid invasion-based methods such as WIP, MIP, HEP, freezing–thaw nuclear magnetic resonance (NMR), and gas physisorption (N_2_, CO_2_, and water vapor), and spontaneous imbibition; (ii) radiation-based methods such as SEM, focus ion beam-SEM (FIB-SEM), micro/nano X-ray and neutron CT, (U)SANS, and (U)SAXS. The different methods can lead to different results for the same sample. The fluid invasion-based methods can only be used to investigate the edge-accessible pores. When conducting experiments, the fluid probes are expected to fully occupy the pore spaces in a short duration (hours to a few days). However, this is often impossible to achieve in tight and low-pore connectivity samples such as shale. For example, in the spontaneous imbibition test, sandstones are saturated with fluid in minutes [1], but it will take 1–15 days to saturate a shale sample [44,45]. Hence, in the fluid-based method, the average pathway length for fluid invasion from the sample surface to the interior pores becomes longer with increasing sample size. Within a reasonably short experimental duration, pores will not always be accessible and occupied by fluid. In contrast, the radiation-based methods have the advantage of investigating both fluid-accessible and -inaccessible pores. However, generating results with those methods is usually either expensive and/or time-consuming (e.g., micro/nano X-ray CT and FIB-SEM) or the experimental resources are not readily available (e.g., neutron CT, (U)SANS, and (U)SAXS). In addition, most of the radiation-based methods suffer a balance of measurable pore size and sample size; for example, 3D μm CT is limited to μm sized pore diameters for mm sized samples, while nm-CT and FIB-SEM can only cover >65 nm pores in 65 μm samples and >10 nm pores for ~10 μm cubed volume [46].

When mercury invades the pores of those tight rocks, it cannot fully access the pore space due to pores’ poor accessibility and the detected pore volume will be less than the total (both accessible and inaccessible) pore volume. Giesche [47] pointed out that the hysteresis behavior of mercury intrusion and extrusion can reflect the mercury-to-pore accessibility. Figure 8 shows the hysteresis curve of the Woodbine Sandstone and Eagle Ford B samples, and the high mercury-to-pore accessibility for Woodbine Sandstone gives rise to flat hysteresis curves, and the extrusion efficiency of all sample sizes is greater than 80%. The low pore connectivity sample of the Eagle Ford B sample shows that the area between the hysteresis curves gradually decreases with a decreasing sample size and accordingly the extrusion efficiency increases from 6.82% to 89.37%. Figure 9 shows the mercury extrusion efficiency of all other samples, indicating that only Woodbine Sandstone has high pore accessibility in all the tested sample sizes whereas the pore accessibility gradually increases with a decreasing sample size for the other eight samples. In addition, the particle density determined from HEP is a good indicator of the fluid-to-pore accessibility of differently sized samples. It has the advantage of a short test duration and being precise, economical, and easy to operate compared to MIP. A combined HEP method for particle density and the μm sized enveloping method for bulk density was recently published by Zhao et al. [30] in order to investigate the sample size-dependent edge-accessible porosity and associated pore connectivity of rocks. The results show the particle density of Woodbine Sandstone does not change with a decrease in sample size (Figure 4), indicating that all the sample sizes of Woodbine sandstone have similar and high fluid-to-pore accessibility. In contrast, other rocks showing decreasing particle densities with smaller sample sizes indicate that their fluid-to-pore accessibility is lower (Figure 4).

### 4.3. A Sample Size-Dependent Concept of Fluid-to-Pore Accessibility and Its Application to Laboratory Tests

The ratio of edge-accessible to -inaccessible pores reflects fluid flow-to-pore accessibility in rocks (especially low porosity and extremely low permeability rocks) [48,49,50]. However, the sample size effect on the ratio of edge-accessible to -inaccessible pores is not well understood [8]. Here, we employ the concept of the surface zone and interior zone from Tokunaga and Wan [51], which assumes that fluid imbibition first occurs on the rock surface, and then enters into the interior of the rock. The surface zone is defined as the layer under the rock’s outer surface that has a significantly higher fluid accessibility than the interior zone. Dong et al. [52] used artificial sandstone and mercury intrusion to try to explain the sample size effect with a simulation work based on percolation theory and their conclusions supported fluid invasion depth is an influencing factor that controls the sample size effect. Figure 10 illustrates the concept of the surface zone and interior zone distribution in two rocks with different fluid-to-pore accessibilities. In the high accessibility rock, the surface zone and interior zone overlap with each other, and the accessibility of the whole rock is even. No matter the sample size, they show similar porosity and permeability until the sample size is smaller than its representative elementary volume (REV). In the low pore-accessibility rock, when sample size decreases, the thickness of the surface zone will not change, whereas the thickness of the interior zone will gradually shrink. Eventually, when the sample size is equal to twice the surface zone’s thickness, the interior zone disappears, and the sample only has surface zone properties. Due to the limitation of methods used in this study, the thicknesses of the surface zone and interior zone are not able to be determined. To obtain a better knowledge of the REV of different natural rocks and its controlling factors, future works such as Wood’s metal impregnation and tracer tests, followed with laser ablation-inductively coupled plasma-mass spectrometry and contrast-matching (U)SANS, can be utilized to determine the REV [8,42], in addition to a combined particle and bulk density approach [30].

### 4.4. Some Considerations of Representative Sample Size

This work has pointed out that the sample size is a key control of certain petrophysical property measurements. In terms of the representativeness of the rock formation, larger and more intact samples are always better than smaller and crushed samples. The limitation of laboratory-scale experimental capacity, however, often restricts sample sizes to centimeters or less. On the one hand, collecting cores from wells is costly, and operators often only collect rock cuttings with sizes less than 1.7 mm of millimeter sizes. On the other hand, larger samples take a longer time to analyze or sometimes they are too large to be analyzed. The conflict between acquiring robust, representative results and the realities of sample size and laboratory analyses requires a compromise until the current methods are improved and new methods developed. At the very least, researchers need to be aware of the limitations of each technique and find a reasonable sample size.

This study suggests that the most representative values of petrophysical properties require centimeter-sized samples such as cylindrical core plugs, cubes, and irregular rock chips from the core or sidewall core. Whenever such samples are not available, the 1.70–2.38 mm crushed samples or 0.841–1.70 mm cuttings can be considered as a replacement for porosity, pore-throat diameter, and pore diameter characterization when using both fluid- and radiation-based methods. For the techniques which only access pores in the nm range, such as N_2_, CO_2_, and water vapor physisorption, using a crushed sample would not be a serious problem because the pore structure for pores with diameters less than 1000 nm will not be altered, as shown from the (U)SAXS method for the total (accessible and inaccessible pores). However, the fluid accessibility difference between large and small sample sizes will still lead to differences in test results.

Natural rocks are complex and heterogeneous geomaterials. A better understanding of the sample size effect on the petrophysical characterization and fluid-to-pore accessibility of natural rock can help other researchers who study membranes, molecular sieves, concrete, and other porous media to understand pore structure, fluid–solid interaction, and fluid flow at the scale from mm to nm.

## 5. Conclusions

This study investigates the sample size effect on petrophysical property characterization from five different techniques using both fluid- and radiation-based methods. Overall, the sample size effect on the measurement of petrophysical properties can be summarized as (1) physical properties change during the sample crushing process and (2) there is a fluid accessibility difference between large and small sample sizes.

## Figures and Tables

**Figure 1 nanomaterials-13-01651-f001:**
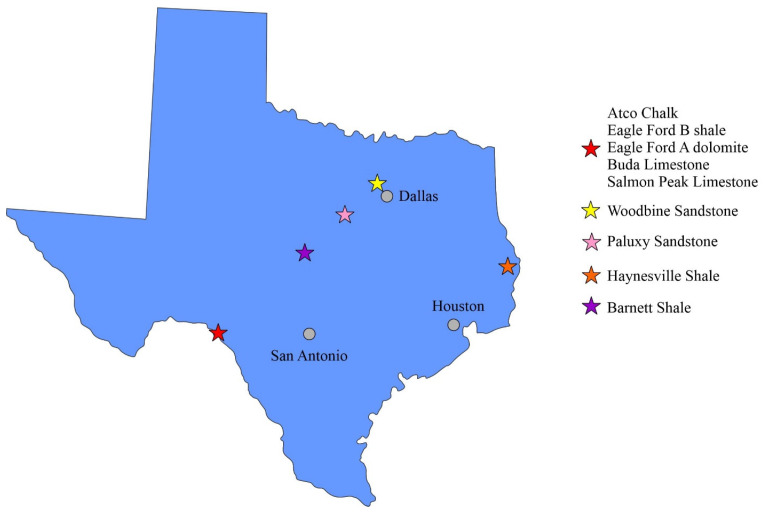
Sample collection locations in Texas.

**Figure 2 nanomaterials-13-01651-f002:**
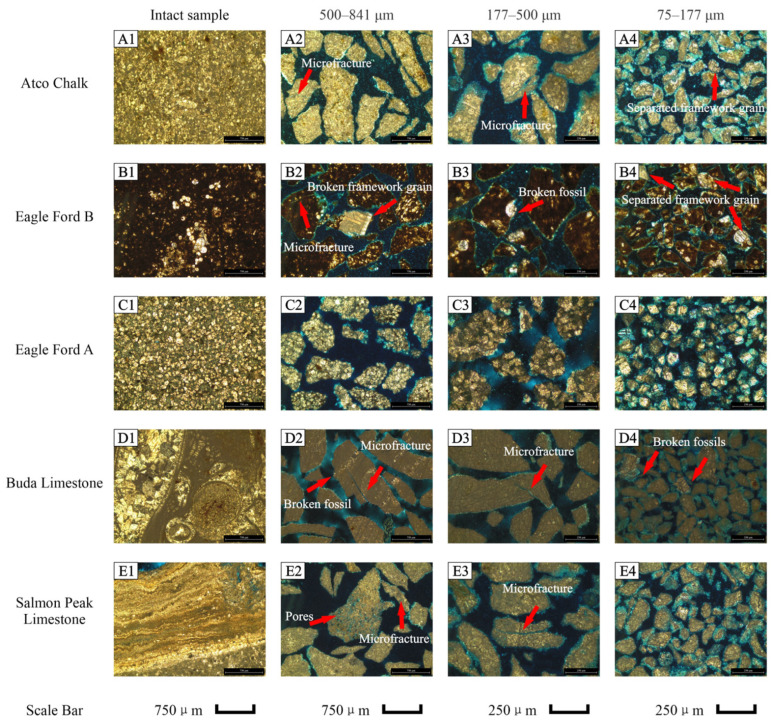
Thin section petrography of intact and crushed samples ((**A1**–**A4**) Atco Chalk; (**B1**–**B4**) Eagle Ford B; (**C1**–**C4**) Eagle Ford A; (**D1**–**D4**) Buda Limestone; (**E1**–**E4**) Salmon Peak Limestone).

**Figure 3 nanomaterials-13-01651-f003:**
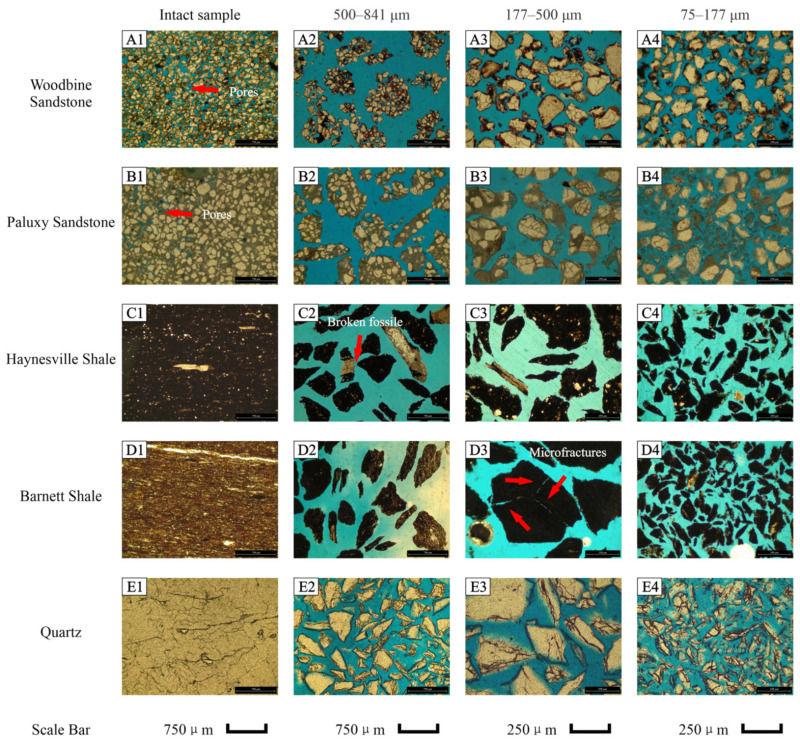
Thin section petrography of intact and crushed samples ((**A1**–**A4**) Woodbine Sandstone; (**B1**–**B4**) Paluxy Sandstone; (**C1**–**C4**) Haynesville Shale; (**D1**–**D4**) Barnett Shale; (**E1**–**E4**) Quartz).

**Figure 4 nanomaterials-13-01651-f004:**
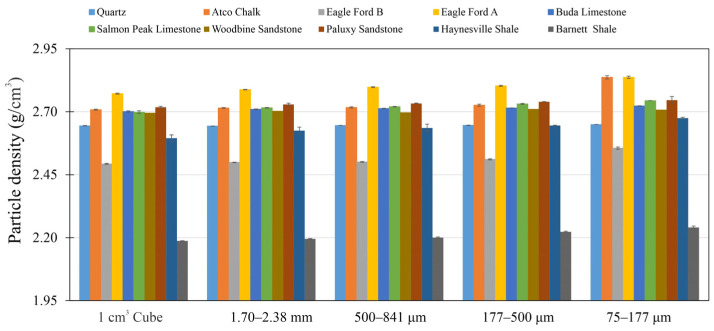
Particle density results from HEP tests of samples with different sizes.

**Figure 5 nanomaterials-13-01651-f005:**
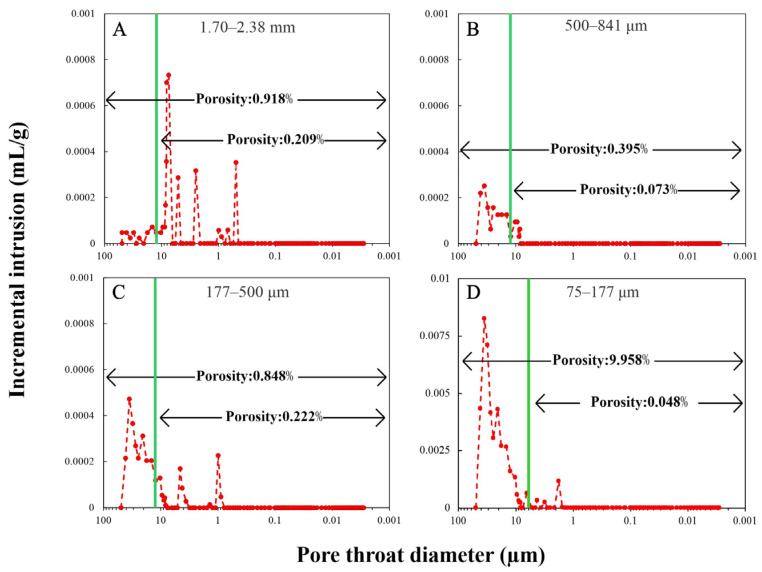
Conformance correction for crushed quartz in MIP tests to derive the inter- and intraparticle porosities. (**A**) 1.70–2.38 mm with conformance pressure at 20 psi. (**B**) 500–841 μm with conformance pressure at 20 psi. (**C**) 177–500 μm with conformance pressure at 20 psi. (**D**) 75–177 μm with conformance pressure at 40 psi. Green line: conformance correction pressure.

**Figure 6 nanomaterials-13-01651-f006:**
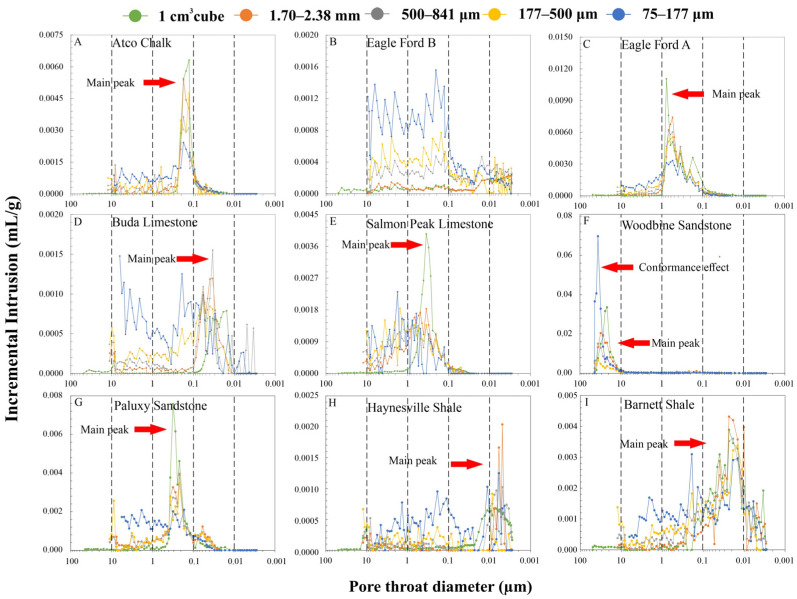
Incremental intrusion vs. pore-throat diameter distribution from MIP tests of five different sizes for nine samples. (**A**–**I**) Atco chalk, Eagle Ford B, Eagle Ford A, Buda Limestone, Salmon Peak Limestone, Woodbine Sandstone, Paluxy Sandstone, Haynesville Shale, and Barnett Shale.

**Figure 7 nanomaterials-13-01651-f007:**
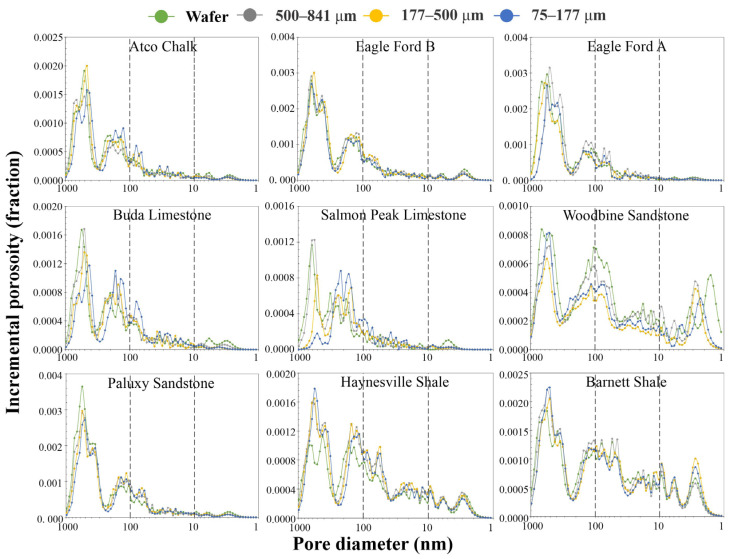
Incremental porosity (fraction) vs. pore diameter distribution from (U)SAXS tests of five different sizes for nine samples.

**Figure 8 nanomaterials-13-01651-f008:**
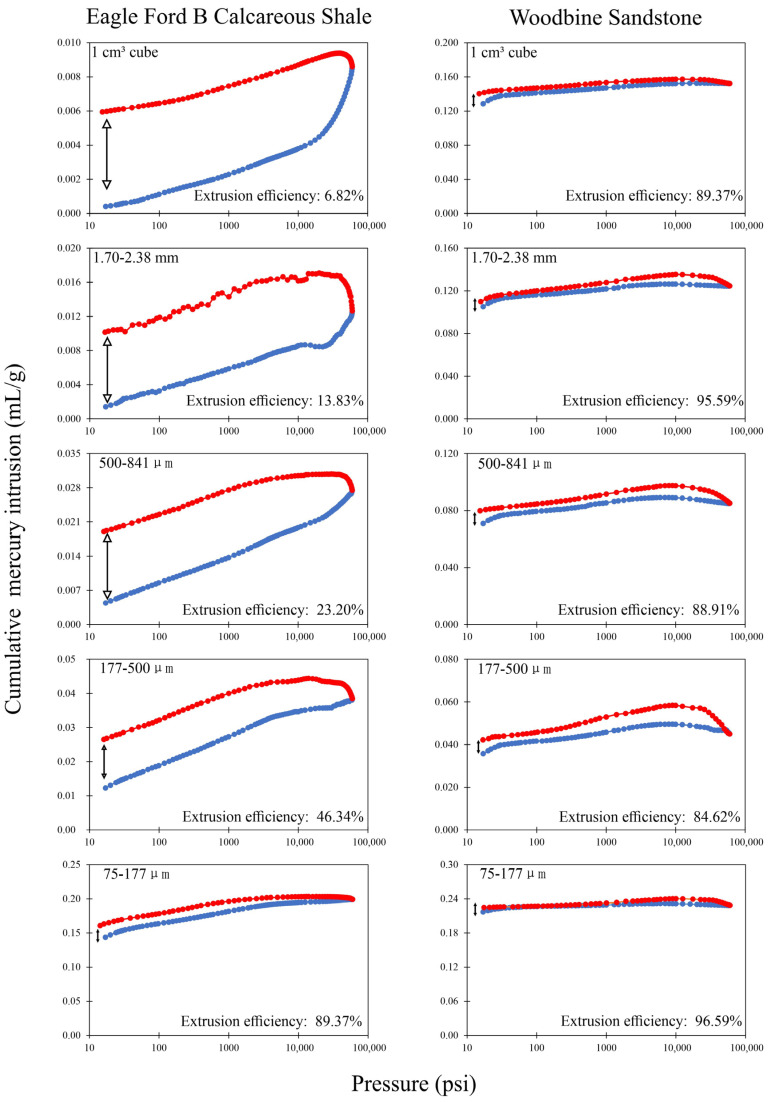
Mercury intrusion–extrusion hysteresis of Eagle Ford B Calcareous Shale and Woodbine Sandstone. Blue dots: mercury intrusion. Red dots: mercury extrusion.

**Figure 9 nanomaterials-13-01651-f009:**
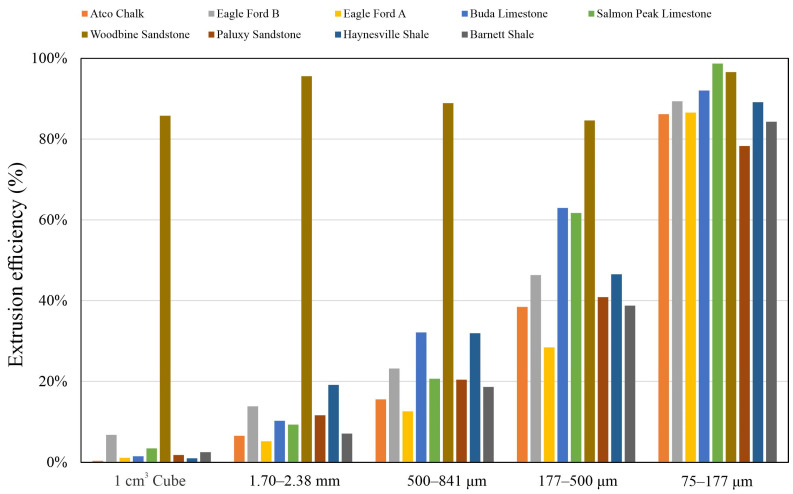
Mercury extrusion efficiency vs. sample sizes.

**Figure 10 nanomaterials-13-01651-f010:**
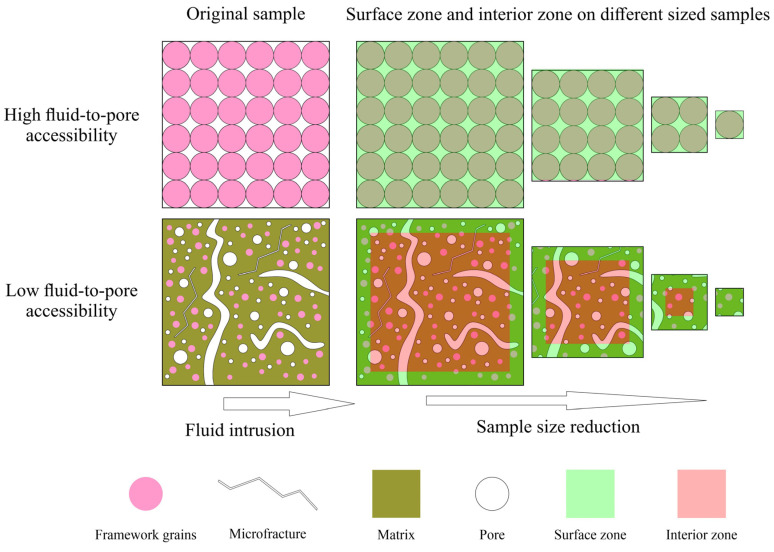
Implication of surface zone/interior zone for two different fluid-to-pore accessibilities.

**Table 1 nanomaterials-13-01651-t001:** Sample size used for each testing method.

Methods	Sample Sizes								
	Intact Samples				Crushed Samples				
	3 cm Cubed	2 cm Cubed	1 cm Cubed	0.8 mm Thick Wafer	1.70–2.38 mm	500–841 μm	177–500 μm	75–177 μm	<75 μm
XRD									√
Petrography		√				√	√	√	
HEP			√		√	√	√	√	
WIP	√	√	√						
MIP			√		√	√	√	√	
(U)SAXS				√		√	√	√	

note: XRD: X-ray diffraction; HEP: helium expansion pycnometry; WIP: water immersion porosimetry; MIP: mercury intrusion porosimetry; (U)SAXS: (ultra-) small-angle X-ray scattering.

**Table 2 nanomaterials-13-01651-t002:** Mineral composition (wt.%) of nine rock samples.

Sample ID	Age	Lithology	Mineral Composition (wt.%)												
			Quartz	Orthoclase	Albite	Anorthite	Calcite	Ankerite	Kutnohorite	Pyrite	Magnetite	Goethite	Fluorapatite	Ulvospine	Clays
Atco Chalk	Late Cretaceous	Limestone	1.0				99.0								
Eagle Ford B Calcareous Shale	Late Cretaceous	Shale	15.5				79.6	1.2		0.8					2.9
Eagle Ford A Dolomitic Ash Bed	Late Cretaceous	Dolomite	9.8				0.7	44.8	36.5		0.2	2.0			6.0
Buda Limestone	Late Cretaceous	Limestone	1.3				98.7								
Salmon Peak Limestone	Early Cretaceous	Limestone	0.2				99.8								
Woodbine Sandstone	Late Cretaceous	Sandstone	91.8									8.2			
Paluxy Sandstone	Early Cretaceous	Sandstone	42.5					57.5							
Haynesville Shale	Late Jurassic	Shale	25.1		2.7	1.0	48.3			5.6				1.5	15.8
Barnett Shale	Mississippian	Shale	40.7	1.0	1.3		2.2						13.1		41.7

**Table 3 nanomaterials-13-01651-t003:** Porosity (%) comparison of different sample sizes between WIP and MIP methods.

Sample	WIP Porosity (%)			Conformance Correction	MIP Porosity (%)				
	3 cm Cubed	2 cm Cubed	1 cm Cubed		1 cm Cubed	1.70–2.38 mm	500–841 μm	177–500 μm	75–177 μm
Atco Chalk	7.92	7.38 ± 0.33	8.50 ± 0.60	Before	7.73	8.02	7.28	9.79	35.69
				After		7.64	6.40	7.46	9.14
Eagle Ford B Calcareous Shale	4.27 ± 0.11	1.64 ± 0.16	1.37 ± 0.38	Before	2.05	3.25	6.51	8.99	33.01
				After		2.85	5.43	7.54	9.67
Eagle ford A Dolomitic Ash Bed	15.10 ± 0.32	15.49 ± 0.06	15.92 ± 0.19	Before	15.39	16.04	15.92	15.43	39.04
				After		15.72	15.09	13.46	13.00
Buda Limestone	3.53 ± 0.10	3.468 ± 0.30	3.70 ± 0.26	Before	3.17	2.89	4.27	6.50	40.60
				After		2.56	3.23	3.72	6.77
Salmon Peak Limestone	7.16 ± 0.16	7.74 ± 1.00	8.21 ± 0.91	Before	7.20	8.00	9.28	12.62	25.38
				After		7.47	8.06	8.59	5.79
Woodbine Sandstone	32.75 ± 0.29	32.33 ± 0.24	31.68 ± 0.17	Before	28.22	22.97	22.25	10.05	35.28
				After					1.66
Paluxy Sandstone	11.08 ± 0.09	10.81 ± 0.70	10.97 ± 1.00	Before	10.01	9.27	9.07	11.73	34.98
				After		8.92	7.73	8.44	11.30
Haynesville Shale	4.73	5.36 ± 0.82	5.46 ± 0.48	Before	4.26	2.48	4.15	5.26	38.86
				After		2.17	2.84	2.27	7.06
Barnett Shale	14.38 ± 0.53	16.38 ± 0.53	17.2 ± 0.44	Before	13.14	13.82	13.81	13.86	47.23
				After		12.39	12.61	9.81	12.03

## Data Availability

The data presented in this study are available on request from the corresponding author.

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
