# Peer review of "Sample Size Effects on Petrophysical Characterization and Fluid-to-Pore Accessibility of Natural Rocks"

_nanomaterials, 2023, doi:10.3390/nano13101651_

Round 1
Reviewer 1 Report
The paper titled "Sample size effects on petrophysical characterization and fluid-to-pore accessibility of natural rocks" provides insights into the impact of sample size on the petrophysical properties of natural rocks. The authors have systematically investigated the effects of sample size on different petrophysical properties using a combination of laboratory techniques. I woud recommend this paper for publication after taking into account the following comments which aim to evaluate the paper and provide recommendations based on its findings:
1- Provide a clearer statement of the research problem: While the introduction provides a good overview of the challenges associated with laboratory-scale petrophysical analyses of natural rocks, it could benefit from a clearer statement of the specific research problem being addressed in this study.
2- Expand on the significance of the research: While the introduction briefly mentions the potential applications of petrophysical analyses in various fields, it could benefit from a more detailed discussion of the potential impact of the research on these fields and the broader scientific community.
3- Provide more information on the study area: The introduction briefly mentions that the samples were collected from Texas, but it does not provide any information on the specific location or geological context of the samples. It would be helpful to provide more information on the geological history and characteristics of the study area.
4- Use simpler language: The introduction contains many technical terms and jargon that may be difficult for a non-specialist to understand. It would be helpful to use simpler language and provide more explanations of technical terms and concepts.
5- Provide a clearer statement of the research objectives: While the introduction provides a good overview of the methods used in the study, it could benefit from a clearer statement of the specific research objectives and hypotheses being tested. This would help readers better understand the purpose and significance of the research.
Author Response
Reply to Review 1:
The paper titled "Sample size effects on petrophysical characterization and fluid-to-pore accessibility of natural rocks" provides insights into the impact of sample size on the petrophysical properties of natural rocks. The authors have systematically investigated the effects of sample size on different petrophysical properties using a combination of laboratory techniques. I would recommend this paper for publication after taking into account the following comments which aim to evaluate the paper and provide recommendations based on its findings:
Thank you very much for your efforts for reviewing this paper. Your valuable suggestions and comments are carefully read and revised accordingly.
1- Provide a clearer statement of the research problem: While the introduction provides a good overview of the challenges associated with laboratory-scale petrophysical analyses of natural rocks, it could benefit from a clearer statement of the specific research problem being addressed in this study.
Thanks for the suggestion. The statement is added to the introduction part as” the influences of the crushing process and sample size effect on petrophysical test results have been noticed but have not been synthetically investigated yet. It is critical to investigate these unknowns to improve the representativeness of, and confidence in, data from laboratory analyses of various rock types”.
2- Expand on the significance of the research: While the introduction briefly mentions the potential applications of petrophysical analyses in various fields, it could benefit from a more detailed discussion of the potential impact of the research on these fields and the broader scientific community.
This study is aiming the natural rocks which is the major target of geo-energy field such as petroleum recovery, coal mining, CO2 sequestration and utilization, nuclear waste reposi-tory, groundwater protection, and geothermal energy exploration. This study not only restricted to rocks but also provides valuable experience for other porous media researches. Therefore, we added a paragraph into the discussion section to extend the impact of this study. “Natural rocks are complex and heterogeneous geo-materials. A better understanding of the sample size effect on petrophysical characterization and fluid-to-pore accessibility of natural rock can help other researchers who study membranes, molecular sieves, con-crete, and other porous media understand pore structure, fluid-solid interaction, fluid flow at the scale from mm to nm.”
3- Provide more information on the study area: The introduction briefly mentions that the samples were collected from Texas, but it does not provide any information on the specific location or geological context of the samples. It would be helpful to provide more information on the geological history and characteristics of the study area.
Thanks for the comments. We carefully considered your suggestion about adding more information about samples. In this paper, we are more care about the rock type such as shale. Limestone, and sandstone, rather than one specific rock. Research object can be any similar rock type from other formations and locations. We think adding extra sample information is less important and will make this paper too long.
4- Use simpler language: The introduction contains many technical terms and jargon that may be difficult for a non-specialist to understand. It would be helpful to use simpler language and provide more explanations of technical terms and concepts.
Revision has been made according to the suggestion. Term sample size effect is explained as “test results variation lead by different sample sizes” and petrophysical properties is explained as “porosity, pore throat distribution, pore diameter distribution, and connectivity” in this study.
5- Provide a clearer statement of the research objectives: While the introduction provides a good overview of the methods used in the study, it could benefit from a clearer statement of the specific research objectives and hypotheses being tested. This would help readers better understand the purpose and significance of the research.
Thanks, we provided a clear statement in the introduction to better claim our research objective. The statement is added to the introduction part as” the influences of the crushing process and sample size effect on petrophysical test results have been noticed but have not been synthetically investigated yet. It is critical to investigate these unknowns to improve the representativeness of, and confidence in, data from laboratory analyses of various rock types”.
Reviewer 2 Report
The authors presented an interesting paper. The paper is well-written and can be useful for practice. The research design, questions, hypotheses, and methods are clearly stated. The paper is suitable for the journal Nanomaterials.
Minor review:
1. Please ensure the abstract is short but reflects the approach, results, and conclusions correctly and concisely.
2. All abbreviations in the tables should be properly explained in the text. SI units must also be reported for all variables.
3. Please check the readability of images Fig.5 - Fig.8.
4. In references, the DOI should be added for individual sources if possible.
5. In discussion, authors should discuss the results and how they can be interpreted from the perspective of previous studies and the working hypotheses. Future research directions may also be highlighted.
I suggest accepting the paper, but my comments should be resolved.
Author Response
Reply to Review2:
The authors presented an interesting paper. The paper is well-written and can be useful for practice. The research design, questions, hypotheses, and methods are clearly stated. The paper is suitable for the journal Nanomaterials.
Minor review:
Thank you very much for your efforts for reviewing this paper. Your valuable suggestions and comments are carefully read and revised accordingly.
- Please ensure the abstract is short but reflects the approach, results, and conclusions correctly and concisely.
The abstract has been revised to be correctly and concisely. Thanks!
- All abbreviations in the tables should be properly explained in the text. Sl units must also be reported for all variables.
All abbreviations have been noted under the table and all the unit in the table are checked to be SI units.
3.Please check the readability of images Fig.5-Fig.8.
Thanks for the suggestion, in the system we uploaded high quality pictures. Under the layout which automatically generated by the system, seems all the pics are squeezed and the readability of Fig. 5-Fig. 8 is pretty low. We will contact the editor to fix this problem in the system.
- In references, the DOI should be added for individual sources if possible.
We revised all the reference to meet the requirement of Nanomaterials and the DOI are added to each paper. Thanks.
5.In discussion, authors should discuss the results and how they can be interpreted from the perspective of previous studies and the working hypotheses. Future research directions may also be highlighted.
Thanks for your suggestion, we have summarized previous work in the introduction and discussed them in the Chapter 4.1 and 4.2. They only focused on the pore structure changes during the crushing process When people test the pore structure, the most common method is fluid based method. They didn’t notice the influence of sample size effect is co-controlled by the sample’s property and fluid-to-pore connectivity. Therefore, we emphasized more on the discussion of how fluid-to-pore connectivity influence the measurement results in Chapter 4.3.
Future research directions have been highlighted at the end of Chapter 4.3.